# Bayesian Optimization over Bounded Domains with Beta Product Kernels

**Huy Hoang Nguyen**[†]    **Han Zhou**[‡]    **Matthew B. Blaschko**[‡*]    **Aleksei Tiulpin**[†*]

[†]Research Unit of Health Sciences and Technology, University of Oulu, Oulu, Finland
[‡]Dept. ESAT, Center for Processing Speech and Images, KU Leuven, Leuven, Belgium
{huy.nguyen,aleksei.tiulpin}@oulu.fi    {han.zhou,matthew.blaschko}@esat.kuleuven.be

## Abstract

Bayesian Optimization with Gaussian Process (GP) and Matérn and Radial Basis Function (RBF) covariance functions is commonly used to optimize black-box functions. The Matérn and the RBF kernels do not make any assumptions about the domain of the function, which may limit their applicability in bounded domains. To address the limitation issue, we introduce a non-stationary Beta Unit Hyper-Cube (BUC) kernel, which is induced by a product of Beta distribution density functions, and allows to model functions on bounded domains. To provide theoretical insights, we provide analyses of information gain and cumulative regret bounds when using the GP Upper Confidence Bound (GP-UCB) algorithm with the BUC kernel. Our experiments show that the BUC kernel consistently outperforms the well-known Matérn and RBF kernels in different problems, including synthetic function optimization and the compression of vision and language models.

## 1 Introduction

Bayesian Optimization (BO) is a theoretically grounded strategy for sequential optimization of noisy and black-box functions with costly evaluations. A surrogate model, which approximates the objective function and guides the search process is at the core of BO. A Gaussian Process (GP) is frequently employed here due to its flexibility, capability to quantify uncertainty, and inject priors through covariance functions (interchangeably called kernels). A GP $f \sim \text{GP}(\mu(\mathbf{x}), K(\mathbf{x}, \mathbf{x}'))$ is specified by its mean $\mu(\mathbf{x}) = \mathbb{E}[f(\mathbf{x})]$ and covariance function $K(\mathbf{x}, \mathbf{x}') = \mathbb{E}[(f(\mathbf{x}) - \mu(\mathbf{x}))(f(\mathbf{x}') - \mu(\mathbf{x}'))]$. The choice of kernel plays a critical role in encoding prior beliefs about the characteristics of the target function. Among various options, Matérn and Radical Basis Function (RBF) are frequently chosen kernels for a wide range of optimization problems due to their flexibility (Pedregosa et al., 2011; Head et al., 2018; Gardner et al., 2018) [1] . The Matérn kernel is expressed as (Rasmussen, 2003)

$$K_{\text{Matérn}}(r) = \frac{2^{1-\nu}}{\Gamma(\nu)} \left( \sqrt{2\nu} \frac{r}{\ell} \right)^{\nu} K_{\nu} \left( \sqrt{2\nu} \frac{r}{\ell} \right), \tag{1}$$

where $r = \|\mathbf{x} - \mathbf{x}'\|_2$, $\nu > 0$ is a smoothness parameter, $\ell$ is a positive length scale, $\Gamma(\cdot)$ is the Gamma function, and $K_{\nu}$ is a modified Bessel function (Abramowitz and Stegun, 1968). When $\nu \to \infty$, the Matérn kernel is equivalent to the RBF kernel, formulated as

$$K_{\text{RBF}}(r) = \exp \left( -\frac{r^2}{2\ell^2} \right). \tag{2}$$

---

[*]Equal last author
[1]Matérn is the default kernel in various GP libraries: Scikit-optimize, GPyTorch, and GPyOpt.

Workshop on Bayesian Decision-making and Uncertainty, 38th Conference on Neural Information Processing Systems (NeurIPS 2024).

The Matérn and RBF kernels are defined on unbounded domains; however, in many practical applications, a function of choice may be defined on a bounded domain, which may result in sub-optimal performance.

In this work, we propose a novel non-stationary kernel BUC, named after Beta distribution-based Unit Hyper-Cube, which is specifically designed for modeling functions over bounded domains. Our kernel is constructed from a product of multiple Beta distribution density functions, each of which naturally represents a wide range of functions defined on $[0, 1]$. We theoretically derive bounds for the maximum information gain and the cumulative regret when optimizing with the GP Upper Confidence Bound (GP-UCB) algorithm. Our results show that BUC *consistently* outperforms the RBF and Matérn kernels across various tasks, including synthetic function optimization and deep learning-based vision and language model compression.

## 2 Beta Distribution-based Unit HyperCube Kernel

### 2.1 Definition

Let $\mathbf{x}, \mathbf{x}' \in [0, 1]^d$ denote two random variables in a $d$-dimensional unit hypercube. We aim to develop some positive semi-definite function $K : [0, 1]^d \times [0, 1]^d \to \mathbb{R}$. For that purpose, we introduce a function $\phi : [0, 1]^d \to ([0, 1]^d \to \mathbb{R})$, that is

$$\phi(\mathbf{x}, \mathbf{s}) = \prod_{i=1}^{d} \text{Beta}_{h_i}(x_i; s_i) = \prod_{i=1}^{d} \frac{\Gamma(\alpha_i + \beta_i)}{\Gamma(\alpha_i)\Gamma(\beta_i)} s_i^{\alpha_i - 1}(1 - s_i)^{\beta_i - 1}, \tag{3}$$

where $x_i$ represents the mode of the $i$-th Beta distribution, $\alpha_i = \frac{x_i}{h_i} + 1$, $\beta_i = \frac{1 - x_i}{h_i} + 1$, $h_i$ is the smoothing bandwidth of the $i$-th dimension, and $\mathbf{s} = [s_1, \ldots, s_d]^\mathsf{T} \in [0, 1]^d$ is the variables of the Beta distributions. Then, the BUC kernel is expressed as

$$K_{\text{BUC}}(\mathbf{x}, \mathbf{x}') = \int_{[0,1]^d} \phi(\mathbf{x}, \mathbf{s})\phi(\mathbf{x}', \mathbf{s})d\mathbf{s} = C \int_{[0,1]} \cdots \int_{[0,1]} \prod_{i=1}^{d} g(s_i)ds_1 \ldots ds_d, \tag{4}$$

where

$$C = \prod_{i=1}^{d} \frac{\Gamma(\alpha_i + \beta_i)}{\Gamma(\alpha_i)\Gamma(\beta_i)} \frac{\Gamma(\alpha_i' + \beta_i')}{\Gamma(\alpha_i')\Gamma(\beta_i')} \qquad \text{and} \qquad g(s_i) = s_i^{\alpha_i + \alpha_i' - 2}(1 - s_i)^{\beta_i + \beta_i' - 2}. \tag{5}$$

Based on the cumulative distribution function of the Beta distribution $\text{Beta}(\alpha, \beta)$, we have that

$$\int_{[0,1]} s^{\alpha - 1}(1 - s)^{\beta - 1}ds = \frac{\Gamma(\alpha)\Gamma(\beta)}{\Gamma(\alpha + \beta)}, \tag{6}$$

each individual integral becomes

$$\int_{[0,1]} g(s_i)ds_i = \frac{\Gamma(\alpha_i + \alpha_i' - 1)\Gamma(\beta_i + \beta_i' - 1)}{\Gamma(\alpha_i + \alpha_i' + \beta_i + \beta_i' - 2)}. \tag{7}$$

Assume that $s_i$'s are independent of each other, the BUC kernel can be simplified to

$$K_{\text{BUC}}(\mathbf{x}, \mathbf{x}') = C \prod_{i=1}^{d} \frac{\Gamma(\alpha_i + \alpha_i' - 1)\Gamma(\beta_i + \beta_i' - 1)}{\Gamma(\alpha_i + \alpha_i' + \beta_i + \beta_i' - 2)} \tag{8}$$

As $\alpha_i + \beta_i = \alpha_i' + \beta_i' = \frac{1}{h_i} + 2$, the kernel can be expressed as

$$K_{\text{BUC}}(\mathbf{x}, \mathbf{x}') = \underbrace{\prod_{i=1}^{d} \frac{\Gamma^2\left(\frac{1}{h_i} + 2\right)}{\Gamma\left(\frac{2}{h_i} + 2\right)}}_{\tilde{C}(h_1, \ldots, h_d)} \prod_{i=1}^{d} \frac{\Gamma(\alpha_i + \alpha_i' - 1)\Gamma(\beta_i + \beta_i' - 1)}{\Gamma(\alpha_i)\Gamma(\beta_i)\Gamma(\alpha_i')\Gamma(\beta_i')}. \tag{9}$$

We graphically compare the BUC kernel to the Matérn kernel on the unit 1D domain in Figures 1a and 1b. Whereas Matérn is a stationary kernel with a constant diagonal, our proposed kernel is non-stationary.

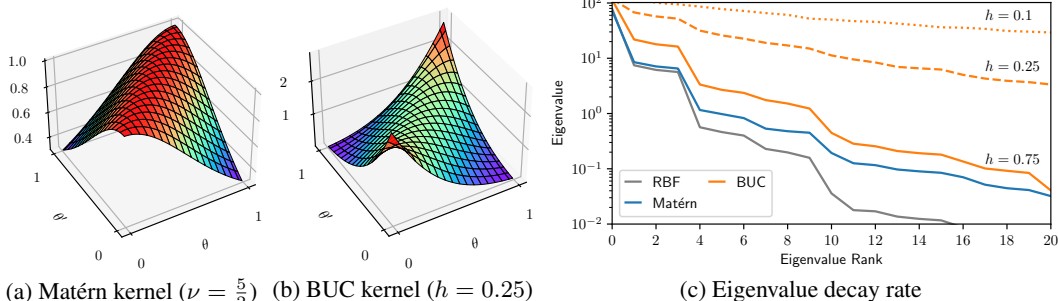

(a) Matérn kernel ($\nu = \frac{5}{2}$)  (b) BUC kernel ($h = 0.25$)  (c) Eigenvalue decay rate

Figure 1: (a-b) Covariance matrices of the Matérn kernel and our BUC kernel on the unit 1D domain. (c) Eigenvalue decay rate comparison between the RBF, Matérn, and BUC kernels.

## 2.2 Bounds on Maximum Information Gain and Cumulative Regret

Consider a GP $f \sim \text{GP}(0, K_{\text{BUC}}(\mathbf{x}, \mathbf{x}'))$ on the $d$-dimensional unit hypercube $D = [0, 1]^d$. Assume that we use the Gaussian Process Upper Confidence Bound (GP-UCB) algorithm to find the optimal solution $\mathbf{x}^* = \arg\min_{\mathbf{x} \in [0,1]^d} f(\mathbf{x})$. Here, we aim to derive bounds of the *maximum information gain* $\gamma_T$ and the corresponding *cumulative regret* $R_T$ after $T$ iterations. $\gamma_T$ is defined as

$$\gamma_T = \max_{A \subseteq D:|A|=T} \text{I}(\mathbf{y}_A, \mathbf{f}_A), \tag{10}$$

where $\mathbf{f}_A = [f(\mathbf{x})]_{\mathbf{x} \in A}$, and $\text{I}(\mathbf{y}_A, \mathbf{f}_A)$ is the information gain, defined as

$$\text{I}(\mathbf{y}_A, \mathbf{f}_A) = \text{H}(\mathbf{y}_A) - \text{H}(\mathbf{y}_A \mid \mathbf{f}_A), \tag{11}$$

where $\text{H}(\cdot)$ is the Shannon entropy. The maximum information gain can also be expressed as $\text{I}(\mathbf{y}_A, \mathbf{f}_A) = \frac{1}{2} \log |\mathbf{I} + \sigma^{-2}\mathbf{K}_A|$, where $\mathbf{K}_A = [K(\mathbf{x}, \mathbf{x}')]_{\mathbf{x}, \mathbf{x}' \in A}$ (Cover, 1999; Srinivas et al., 2009). The cumulative regret is defined as $R_T = \sum_{t=1}^{T} f(\mathbf{x}_t) - f(\mathbf{x}^*)$. We derive the bound of $\gamma_T$ as $\mathcal{O}\left(d \log(T 2^{3d - \frac{2d}{h}} h^{-\frac{3d}{2}})\right)$ in Theorem 1, and the bound of the cumulative regret w.r.t. the BUC kernel in Theorem 2.

**Theorem 1.** *The maximum information gain $\gamma_T$ of the BUC kernel is bounded by* $\mathcal{O}\left(\tilde{h}dT\right)$ *where* $\tilde{h} = \left| 3\log 2 - \frac{2\log 2}{h} + \frac{3}{2}\log\frac{1}{h} \right|$.

*Proof.* We directly bound $\text{I}(\mathbf{y}_A, \mathbf{f}_A) = \frac{1}{2}\log|\mathbf{I} + \sigma^{-2}\mathbf{K}_A| \leq \frac{1}{2}\log|\text{diag}(\mathbf{I} + \sigma^{-2}\mathbf{K}_A)|$ using the Hadamard's inequality. $K_{\text{BUC}}(\mathbf{x}, \mathbf{x})$ can bounded by $2^{3d - \frac{2d}{h}}(\frac{1}{h}+1)^d(\frac{1}{h\pi} + \frac{3}{2\pi})^{\frac{d}{2}}$ given that $\frac{\Gamma^2(x+1)}{\Gamma(2x+1)} = \frac{\sqrt{\pi}\Gamma(x+1)}{2^{2x}\Gamma(x+\frac{1}{2})}$ and $\left(\frac{2}{2x+1}\right)^{\frac{1}{2}} \leq \frac{\Gamma(x+\frac{1}{2})}{\Gamma(x+1)} \leq 2, \forall x \geq 0$. $\qquad\square$

**Theorem 2.** *Let $D = [0,1]^d$ with $d \in \mathbb{N}$, pick $\delta \in (0,1)$, and define $\beta_t = 2\log(t^2 2\pi^2/(3\delta) + 2d\log\left(t^2 dbr\sqrt{\log(4da/\delta)}\right)$. Running the GP-UCB with $\beta_t$ for a sample $f$ of a GP with mean function zero and covariance function $K_{BUC}$, the cumulative regret is bounded as follows*

$$Pr\left\{ R_T \leq T\sqrt{C_1 \beta_T \tilde{h}d} + 2 \; \forall T \geq 1 \right\} \geq 1 - \delta,$$

*where $C_1 = 8/\log(1 + \sigma^2)$.*

*Proof.* Apply Theorem 1 and Theorem 2 of Srinivas et al. (2009). $\qquad\square$

In Figure 1c, we present the spectral decay analysis for the RBF, Matérn, and BUC kernels. Our analysis demonstrates a strong correlation between the bandwidth parameter $h$ and the eigenvalue decay rate of the BUC kernel. With $h < 1$, the BUC kernel shows a slower eigenvalue decay rate

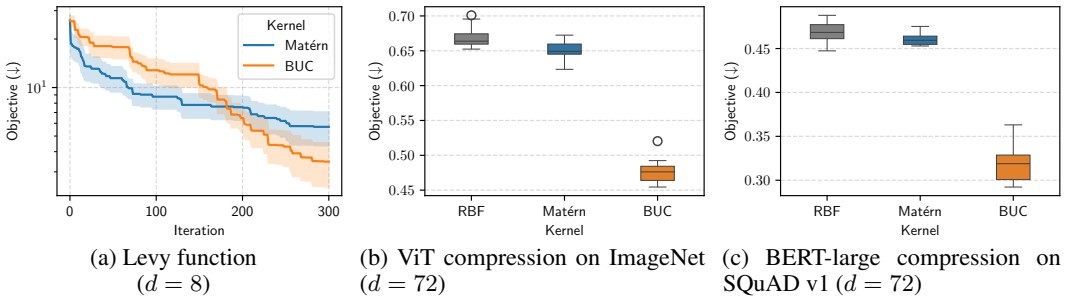

(a) Levy function
$(d = 8)$     (b) ViT compression on ImageNet
$(d = 72)$     (c) BERT-large compression on
SQuAD v1 $(d = 72)$

Figure 2: Comparison between GP using the Matérn kernel ($\nu = 2.5$), and our BUC kernel

Table 1: Quantitative results on vision and language model compression. 'LR' indicates the latency ratio between the original model and its compressed version. The latency measurements were conducted on an Nvidia GTX 2080 Ti.

| (a) ViT compression on ImageNet | | | | (b) BERT-large compression on SQuAD v1 | | | |
| --- | --- | --- | --- | --- | --- | --- | --- |
| Kernel | LR ($\uparrow$) | $\mathcal{R}$ ($\%, \downarrow$) | Acc. ($\%, \uparrow$) | Kernel | LR ($\uparrow$) | $\mathcal{R}$ ($\%, \downarrow$) | F1 ($\%, \uparrow$) |
| RBF | 12.67 | $41.20_{\pm 0.54}$ | $74.26_{\pm 0.02}$ | RBF | 41.08 | $32.84_{\pm 0.52}$ | $86.06_{\pm 0.15}$ |
| Matérn | 14.32 | $39.26_{\pm 0.48}$ | $\mathbf{74.34}_{\pm 0.03}$ | Matérn | 42.17 | $32.18_{\pm 0.21}$ | $\mathbf{86.47}_{\pm 0.06}$ |
| BUC | $\mathbf{14.79}$ | $\mathbf{21.96}_{\pm 0.61}$ | $74.29_{\pm 0.03}$ | BUC | $\mathbf{43.68}$ | $\mathbf{18.09}_{\pm 0.69}$ | $\mathbf{86.44}_{\pm 0.05}$ |

compared to the RBF and Matérn kernels, indicating its superior capacity to capture complex function behaviors on the unit hypercube. In addition, based on Theorem 4 of Srinivas et al. (2009), this slower decay rate suggests that the BUC kernel has a higher upper bound on the maximum information gain than the RBF and Matérn kernels.

## 3 Experiments

**Levy Function** We conducted experiments on the Levy function (Laguna and Marti, 2005). We utilized the GP-UCB algorithm for the minimization. We initially selected 30 data points using the Sobol's algorithm (Sobol', 1967; Owen, 1998), and performed the optimization in 300 iterations. We present the comparisons between the BUC kernel and the Matérn kernel in Figure 2a. In both cases of $d = 4$ and $d = 8$, our kernel converged more slowly during the early iterations but ultimately produced better results.

**Vision and Language Model Compression** We formulated the compression objective as $\min_{\mathbf{x} \in [0,1]^d} \mathcal{L}(\mathbf{x}) + \mathcal{R}(\mathbf{x})$, where $\mathcal{L}(\cdot)$ is the error rate of the compressed model, and $\mathcal{R}(\cdot)$ is the compression rate compared to the original model. We utilized the LoSparse method (Li et al., 2023) to perform low-rank and sparse approximation. We compressed the vision classification model ViT (Dosovitskiy, 2020) on the ImageNet dataset (Deng et al., 2009), and the language model BERT-large (Devlin, 2018) on the SQuAD v1 (Rajpurkar et al., 2016). For both the models, we had $d = 72$. We randomly sampled 5 initial data points, and set $T = 30$. In each iteration, we trained each compressed ViT model for only one epoch, while the compressed BERT-large model was trained for 256 steps (less than one epoch). We repeated the experiments 10 times and reported the results in Figures 2b and 2c. Accordingly, the experiments consistently indicates that the BUC kernel substantially outperformed the two well-known baseline kernels, RBF and Matérn on both tasks. The detailed quantitative results in Table 1 show that our kernel significantly enhanced the compression rate while maintaining minor performance trade-offs.

## 4 Conclusion

We introduce a novel non-stationary kernel, called BUC, tailored specifically for BO on unit hypercubes. We then derive bounds on $\gamma_T$ and $R_T$ for optimization using the GP-UCB algorithm with the

BUC kernel. The experiments show that our kernel consistently outperforms the widely used RBF and Matérn kernels across different optimization tasks.

## Acknowledgments

The authors wish to acknowledge CSC—IT Center for Science, Finland, for generous computational resources. HZ and MBB acknowledge support from the Flemish Government (AI Research Program) and the Research Foundation - Flanders (FWO) through project number G0G2921N. HZ is supported by the China Scholarship Council. A.T. and H.H.N. were supported by the Research Council of Finland (Profi6 336449 funding program) and Sigrid Juselius foundation.

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
