# OpenReview forum: "Bayesian Optimization over Bounded Domains with Beta Product Kernels"
_NeurIPS.cc/2024/Workshop/BDU — NeurIPS BDU Workshop 2024 Poster_

### Official Review · Reviewer_rmRg · 2024-09-16

[review text omitted: it was posted to a different submission]

---

### Official Review · Reviewer_C8DD · 2024-09-28
**Evaluation of the Work: "Bayesian Optimization over Bounded Domains with Beta Product Kernels"**

**Rating:** 6
**Confidence:** 4

**Review:**

The paper presents a new kernel, the Beta Unit Hyper-Cube (BUC) kernel, designed for bounded domains in Bayesian optimization. The quality of the theoretical and experimental work is solid.

The paper is generally clear, especially in its explanation of the motivation for using a bounded-domain kernel. The introduction of a kernel specifically designed for bounded domains is a novel contribution. The BUC kernel addresses an important limitation in Gaussian processes using traditional kernels like Matérn and RBF.

The significance of this work lies in its potential for practical applications in fields requiring optimization over bounded domains, such as hyperparameter tuning in machine learning and model compression.

Pros:
1.Novelty: The BUC kernel fills a gap in Bayesian optimization for bounded domains.
2.Theoretical Rigor: The paper provides a well-justified theoretical framework for the BUC kernel, including bounds on information gain and regret.

Cons:
1.Limited Benchmark Tasks: The experiments are not thorough enough. Expanding the range of real-world benchmarks could further validate the new kernel's robustness.
2.Few Practical Guidelines: While the paper performs well in theory and experiments, it lacks discussion on how to choose parameters like the bandwidth $h$ for the BUC kernel in practice.

---

### Decision · Program_Chairs · 2024-10-09

**Decision:**

Accept (Poster)

**Comment:**

While the scores of this work are borderline accept, the actual text of the reviews is largely positive and makes clear this work is above the bar for a workshop.